# Regulating pharmacists as contraception providers: A qualitative study from Coastal Kenya on injectable contraception provision to youth

Lianne Gonsalves[1,2,3]*, Kaspar Wyss[2,3], Peter Gichangi[4,5,6], Lale Say[1], Adriane Martin Hilber[2,3]

1 Department of Reproductive Health and Research including UNDP/UNFPA/UNICEF/WHO/World Bank Special Programme of Research, Development and Research Training in Human Reproduction (HRP), World Health Organization, Geneva, Switzerland, 2 Swiss Tropical and Public Health Institute (Swiss TPH), Basel, Switzerland, 3 University of Basel, Basel, Switzerland, 4 International Centre for Reproductive Health Kenya, Mombasa, Kenya, 5 Department of Human Anatomy, University of Nairobi, Nairobi, Kenya, 6 Ghent University, Ghent, Belgium

* gonsalvesl@who.int

**Data Availability Statement:** Data cannot be shared publicly because consent procedures for participants did not include making full interview

## Abstract

### Introduction

Young people worldwide are often reticent to access family planning services from public health facilities: instead, they choose to get contraception from private, retail pharmacies. In Kenya, certain contraceptives are available in pharmacies: these include injectables, which can be dispensed but not administered, according national guidelines. However, Kenya struggles with enforcement of its pharmacy regulations and addressing illegal activity. Therefore, in this qualitative study, we assessed private pharmacies as an existing source of injectable contraception for young Kenyans (age 18–24), and investigated the perceived quality of service provision.

### Methods

This study used: focus group discussions (6) with young community members; in-depth interviews (18) with youth who had purchased contraception from pharmacies; key informant interviews with pharmacy personnel and pharmacy stakeholders (25); and a mystery shopper (visiting 45 pharmacies).

### Results

The study found that for injectable contraception, private pharmacies had expanded to service provision, and pharmacy personnel's roles had transcended formal or informal training previously received–young people could both purchase and be injected in many pharmacies. Pharmacies were perceived to lack consistent quality or strong regulation, resulting in young clients, pharmacy personnel, and regulators being concerned about illegal activity. Participants' suggestions to improve pharmacy service quality and regulation compliance

and focus group discussion transcripts publicly available. Excerpts from transcripts are available on request from corresponding author and following approval from the University of Nairobi/Kenyatta National Hospital Ethics Committee (contact via uonknh_erc@uonbi.ac.ke) for researchers who meet the criteria for access to confidential data.

**Funding:** This study was partially supported by the UNDP/UNFPA/UNICEF/WHO/World Bank Special Programme of Research, Development and Research Training in Human Reproduction (HRP). The funders had no role in study design, data collection and analysis, decision to public or preparation of the manuscript.

**Competing interests:** The authors have declared that no competing interests exist.

focused on empowering consumers to demand quality service; strengthening regulatory mechanisms; expanding training opportunities to personnel in private pharmacies; and establishing a quality-based 'brand' for pharmacies.

## Discussion

Kenya's recent commitments to universal health coverage and interest in revising pharmacy policy provide an opportunity to improve pharmacy quality. Multi-pronged initiatives with both public and private partners are needed to improve pharmacy practice, update and enforce regulations, and educate the public. Additionally, the advent of self-administrable injectables present a new possible role for pharmacies, and could offer young clients a clean, discreet place to self-inject, with pharmacy personnel serving as educators and dispensers.

## Introduction

Young people around the world are often reticent to access sexual and reproductive health (SRH) services, including family planning services, from public health facilities–well-established obstacles include provider bias, a lack of privacy, few contraceptive options, limited financial resources, and legal and policy (real or presumed) barriers[1]. Instead, they may seek contraception from non-judgmental, confidential, and convenient sources: retail pharmacies provide one such option[2, 3]. Pharmacies are sources of contraceptive commodities and services[3], particularly condoms, emergency contraception, and daily oral contraceptive pills, all of which can be accessed informally or formally without prescriptions in many countries [4, 5]. Very effective injectable contraceptives[6], including the popular depot medroxyprogesterone acetate (DMPA) are also sold[7].

For both contraceptive and broader health services, the role of pharmacists in low- and middle-income countries (LMICs) has rapidly broadened into that of a service provider. A Cochrane review in 2013 described the health impacts of pharmacist-provided services, finding that they could positively-impact certain clinical outcomes related to management of non-communicable diseases (e.g. management of glucose levels for diabetic clients, as well as management of hypertension and asthma) and reduce visits to healthcare providers[8]. However, enthusiasm for using pharmacies to expand the reach of health services is tempered with apprehension about the quality of services provided: two systematic reviews of pharmacy services in LMICs found common problems in counselling and questioning of clients, inaccurate diagnoses, poor referral, inappropriate medicine sales, and a lack of adherence to prescribing and advising protocols[9, 10].

### Retail pharmacy policy in Kenya

In Kenya, private retail pharmacies (locally referred to as *chemist shops*), are registered, regulated and inspected by the Ministry of Health's specialized agency, the Pharmacy and Poisons Board (PPB)[11]. The PPB is also tasked with licensing pharmacy professionals, and registering and regulating the manufacture and sale of medicines and medical devices [11]. Pharmacy compliance standards are determined by the country's Pharmacy and Poisons Act [11], originally drafted in 1957 when pharmacy activities were traditionally limited to drug supply and dispensing. Private pharmacies can only be legally opened by registered pharmacists (requiring a five-year bachelor's degree in pharmacy or higher) or pharmaceutical technologists

(requiring a two year, post-secondary diploma program) [11]. Once open, they operate independently as for-profit businesses, fully removed from the commodity procurement, reporting, and training infrastructure used in public sector pharmacies. Private pharmacy registrations must be renewed annually [11]. These private pharmacies are the only legally recognized tier of retail drug outlet, making Kenya unique in a region where other countries also accredit lower-tier drug shops[7].

Pharmacists and pharmaceutical technologists must renew their licenses ever year[12]. Throughout their practicing career and as a mechanism for maintaining and building knowledge and skills, pharmacists and pharmaceutical technologists have to complete a certain number of 'continuing professional development' (CPD) credits per year. A number of membership-based professional societies, including the Kenya Pharmaceutical Association (for pharmaceutical technologists) and the Pharmaceutical Society of Kenya (for pharmacists) have traditionally carried out CPD for their respective cadres[13, 14]. That said, CPD programs have been challenged by a limited capacity to regulate and enforce CPD requirements, poor organization, and limited support from pharmacists' employers[14].

## Kenyan policies on contraception provision through pharmacies, including to young people

For contraception specifically, Kenya's 2009 National Family Planning Guidelines, indicate that an array of commodities, including condoms, emergency contraception, daily contraceptive pills, and injectables can be dispensed in pharmacies [15]. Both progestin-only injectables (such as DMPA) as well as combined injectable contraceptives are available in private pharmacies, with a single note adding that the client is to be referred for the injection itself [15].

Young people's ability to access contraception was strongly supported by Kenya's Adolescent Reproductive Health Policy in 2003, which aimed to double the use of modern contraceptives among 15–24 year-old by 2015[16]. The updated 2015 Adolescent Sexual and Reproductive Health Strategy affirmed a commitment to promoting young people's sexual and reproductive health and rights, and acknowledged the need for strategic partnerships between the public and private sector to ensure SRH services were delivered in a manner responsive to adolescents' specific needs and vulnerabilities[17]. That said, there are no specific references to the role that pharmacies play in contraception provision (injectable or otherwise) in these adolescent-specific strategic documents. Additionally, pharmacy practice guidance documents also contain no instructions on pharmacy provision of any contraception [7].

According to the 2014 Kenya Demographic and Health Survey, the unmet need for family planning among young women remains high: 23.0% of currently married 15–19 year-old women and 18.9% of currently married 20–24 year-old women. Among sexually active, unmarried young women, 49.9% of 15–19 year-olds and 30.7% of 20–24 year-olds are not currently using any contraceptive[18]. Short-acting modern family methods available through pharmacies (including injectable contraception) have proven popular among sexually active young women [18], and private pharmacies in Kenya are a relied-upon source of contraception commodities for adults and young people alike [18–20].

It is important to support contraception provision through the existing channels preferred by young people. However, like many countries, Kenya struggles with enforcement of its existing pharmacy regulations and addressing illegal pharmacy activity [7, 21]. Additionally, there has been minimal documentation as to the quality of contraception provision services in Kenyan pharmacies, and we are not aware of any study exploring the quality of injectable contraception provision to young clients in particular. Therefore, in this qualitative study, we investigated private pharmacies as an existing source of injectable contraception for young

Kenyans, and sought to understand the quality of the service provision and perceived influence of the current regulatory climate.

## Methods

This study was embedded in a larger mixed methods study which sought to understand how young people aged 18–24 in Coastal Kenya use their local pharmacies to access contraception. Data collection took place in Kwale County (one of six counties in Kenya's former Coast Province), specifically the peri-urban towns and surroundings of Kwale and Ukunda between November 2017 and March 2018. Table 1 describes all the methods used. Study instruments are included as Supporting Information.

We used several qualitative methods to describe the experience dispensing and administering injectable contraception and other contraceptive methods, both from the perspectives of young people who might purchase it, as well as the pharmacy personnel who provide it. Six focus group discussions (FGDs) were carried out with people aged 18–24, purposively recruited by data collectors from the study area. Three FGDs were conducted with men, and three with women; there were 8–10 participants in each FGD. Additionally, 18 in-depth interviews (IDI) were conducted with young people aged 18–24 who had recently purchased contraception from a pharmacy. IDI participants were recruited in one of two ways: 1) selected pharmacists in the study area handed out leaflets with the study information to young customers following their purchase of any contraception, and; 2) a young data collector spent three evenings stationed outside of two popular pharmacies recruiting young people who had just purchased contraception by describing the study and collecting their contact information.

Data collectors also conducted a full mapping of the pharmacies in the study area, identifying 60 pharmacies in total. Pharmacies (whether open or closed at the time of the visit) were mapped via digital form with an embedded geolocator. From this group, a random sample of pharmacies was generated, and data collectors conducted key informant interviews with one person working at the pharmacy (their training or formal role was not important), until saturation was reached. A total of 19 interviews of pharmacy personnel were conducted. Additionally, six interviews were conducted with representatives of the Ministry of Health, Pharmacy

**Table 1. Description of study methods.**

| | Participant inclusion criteria | Topics addressed | N |
|---|---|---|---|
| Focus Group Discussions | • Age 18–24<br>• Community members | • What happens when young people attempt to access contraception in pharmacies<br>• Feelings about contraception service quality at all access points<br>• Suggestions to improve the regulation and quality of pharmacy services | 6<br>(58 participants) |
| In-depth interviews | • Age 18–24<br>• Recently purchased contraception at pharmacy | • Actual experience accessing injectable or other contraception in private pharmacies<br>• Feelings about quality of contraceptive service provided<br>• Suggestions to improve the regulation and quality of pharmacy services | 18 |
| Key Informant Interviews | • Age 18+<br>• Pharmacy personnel (any role) OR<br>• Pharmacy-related stakeholder (Ministry of Health; regulatory agency; professional association; non-governmental organization) | • Current pharmacy policies in Kenya, especially as they relate to injectable provision for young people<br>• What happens when young people attempt to access contraception in pharmacies<br>• Suggestions to improve the regulation and quality of pharmacy services | 19<br>(pharmacy personnel)<br>6 (stakeholders) |
| Mystery shopper | *Trained youth data collector served as mystery client* | • Actual experience accessing injectable contraception in private pharmacies | 45 visits |

and Poisons Board (PPB), the Pharmaceutical Society of Kenya, and non-governmental organizations working on contraceptive commodity development and provision. These stakeholders worked at either County (Kwale-based), Regional (Mombasa-based), or National (Nairobi-based) levels.

Fifty of the 60 enumerated pharmacies also consented to be visited by a mystery shopper, whose objective was to observe firsthand whether any family planning injection was available for sale in each pharmacy and to inquire whether she could be injected on site. Our mystery shopper was a young data collector who assumed the fictional persona of a typical young Kenyan who would need an injectable contraception. Her persona, developed in close collaboration with young people in the community and our youth data collectors, was: a newly married 24-year-old with no children, who had received her first ever Depo Provera injection from a health clinic three months prior (making her due for another injection). Our mystery shopper successfully visited 45 of the 50 pharmacies.

## Data collection

All qualitative methods used semi-structured guides. FGD, IDI and key-informant interviews were audio-recorded with participants' permission and data was collected in Swahili, English or a mix of the two, based on participant preference. The time and location of interviews were set based on participants' availability and preference. The mystery shopper was provided with short, semi-structured digital forms to complete on mobile phones. The form was adapted from a previous family planning mystery shopper study[22] and included questions about the pharmacy, its staff, and the purchase-related interaction. The mystery shopper was instructed to complete the form immediately after each visit. We were able to determine the registration status of 40 of the 45 pharmacies visited by the mystery shopper using an official list of registered pharmacies in Kwale County, obtained from Kenya's Pharmacy and Poisons Board.

## Analysis

All qualitative data was transcribed and translated (if necessary) into English. The qualitative data were analyzed through an iterative approach to thematic analysis, using inductive and deductive (informed by a research objective to understand dispensing practice) coding on an initial cross-section of transcripts to develop a coding framework. Qualitative analyses were conducted in Atlas.ti Version 8. Quantitative data from the mystery shopper exercise was analyzed in Stata Version 14, and analyses consisted of descriptive statistics and a chi-square test to test any association between a pharmacy being willing to inject our mystery shopper and its registration status. Quantitative and qualitative results were triangulated across methods to validate findings.

## Ethics

Informed consent was obtained from all participants in writing. This study received ethics approval from the Ethikkommission Nordwest- und Zentralschweiz (EKNZ) (Req-2017-00389) in Basel, Switzerland, as well as the University of Nairobi/Kenyatta National Hospital in Nairobi, Kenya (P274/05/2017).

# Results

## Current private pharmacy practice: Can young people obtain injectable contraception?

Participants confirmed that injectable contraceptives were routinely sold and administered in pharmacies. Youth interview participants who reported recently purchasing an injectable

contraceptive indicated that they had also been injected at the pharmacy. The mystery shopper's experience reflected the qualitative findings as well. As can be seen in Table 2, the mystery shopper was told that she would be able to purchase the injection in 29 of the 45 (65%) pharmacy shops she visited; in the 16 (36%) where she could not purchase the injectable, it was because the injectable was either out of stock (15 pharmacies) or was not stocked at all (one pharmacy). In 44% of the visited pharmacies, the mystery shopper was told she could both purchase *and* receive the injection on site. However, in 12 (27%) pharmacies, she was told that the contraceptive injection was not available *and* that the pharmacy did not inject. It should be noted of the 21 (47%) pharmacies where the mystery shopper was told that she could *not* be injected on site, she reported that 12 referred her to either an individual or clinic qualified to provide the injections.

Finally, of the 40 pharmacies visited by our mystery shopper whose registration status we could verify, almost two-thirds (63%) were not appropriately registered. Registration status was not significantly associated with being willing to inject (chi2 = .3274, P = .567).

In alignment with national family planning guidelines, pharmacy stakeholders and personnel all agreed that trained pharmacy personnel could dispense injectables. However, when it came to injectable administering, opinions and rationale diverged. All told, pharmacy stakeholders and personnel identified two ways in which clients purchasing injectable contraception could also be injected at a pharmacy: a qualified medical professional administers the injection (regulation-compliant); or a trained or untrained pharmacy personnel administers the injection (regulation non-compliant).

Some pharmacies (especially larger pharmacies) might be staffed by both a pharmacist/pharmaceutical technician to dispense medication *as well as* another medical professional who was also authorized to administer injections, such as a clinical officer or a nurse. In these scenarios, pharmacies effectively became a private, one-stop-shop for certain health services, including injectable contraception provision.

"The other person who works here is the clinical officer. . .His roles are like those of giving injections. He told me not to give injections to anyone. . .He does that work. . .Other problems like chest problems, he listens to the heart like asthma or pneumonia, things like that. The rest, I am the one who does."–Pharmaceutical technologist

Alternatively, interviews with both pharmacy stakeholders and local pharmacy personnel also revealed that injections were also administered by pharmacists, pharmaceutical technicians and untrained pharmacy workers. Only one untrained pharmacy worker admitted to a data collector that she had been instructed to provide injections herself. Of the two additional interviewees who said that they personally administered injections, in one of these cases the respondent–a nurse–also indicated that she also owned the pharmacy, which she was *not* authorized to do.

One additional, regulation-compliant option was a reciprocal referral system between pharmacies and nearby public or private clinics. Many interviewed pharmacy personnel who said

**Table 2. Ability of mystery shopper to both purchase injectable contraception from the pharmacy and be injected on site, as reported to her by attending pharmacy personnel.**

| | Can be injected on site | | |
|---|---|---|---|
| **Can purchase injectable** | **No (n = 21)** | **Yes (n = 24)** | **Total (n = 45)** |
| No (n = 16) | 27% | 9% | 36% |
| Yes (n = 29) | 20% | 44% | 64% |
| Total (n = 45) | 47% | 53% | 100% |

that they would not inject clients themselves described themselves as having close ties to neighborhood doctors or clinics (usually located very close to the pharmacy).

"Aah I don't inject. There is a clinic in front here where they inject, so most times when they come, we sell them the injectable and she goes to get injected. The referral is just near here."–Pharmacist

Some respondents indicated that they would even offer to call a doctor at the clinic, to let them know a customer was coming. Many interviewed pharmacy personnel added that this relationship was reciprocal, and clinics would send clients to their pharmacy to purchase an injectable contraceptive.

## Perception of quality of private pharmacy service provision

Both pharmacy regulators and young client participants agreed that private pharmacies lacked consistent quality. Participants were aware of the variation in whether and how injectables (and other contraception services) were provided and expressed concern about a lack of regulation among private pharmacies in Kenya. Both groups described an illegal part of the pharmacy sector, which included persons offering health products of doubtful quality (in the worst case, counterfeit); owning and operating a pharmacy without appropriate licensing or training; or dispensing commodities without trainings.

Youth participants indicated that a visit to a new pharmacy in the area risked the purchase of sub-standard commodities from an untrained worker in an unlicensed establishment.

"Those who sell those drugs [in a pharmacy] most of them are not people who are qualified. Most of them are people from school, and being told to go and help there, the person will help. When you go there to buy drugs you find that he/she is selling otherwise and those drugs were needed in another dose."–Youth female community member

Young community members also expressed concern at the government's inability to regulate this illegal activity.

"The government should be keen on licensing these people that open up those chemist, because there are things like medicine, [some] go through the right channel but others go through the back [illegal] door."–Youth male community member

For their part, regulators recognized that their ability to monitor with current resources was limited, and they indicated that they did not feel like they were able to keep pace with illegal pharmacies. One inspector described challenges to conducting mandated periodic inspections of the pharmacies in Kwale County:

"You'll find that these [illegal pharmacies] they've just sprung up suddenly. When they find that we are heading towards that place, they normally close it. And they have . . .these social media network: the person will just text and say that 'PPB, their vehicle was seen somewhere in Kwale'. We move around and find that the illegal outlets are closed. . .so we have to wait for the next day. We try to [show] that we are going back to Mombasa and then come back maybe in the evening and see whether they are open."–Pharmacy and Poisons Board inspector

Regulators and other pharmacy stakeholders, therefore, shared young people's concerns about unqualified pharmacy personnel dispensing counterfeit commodities, including contraception commodities.

> "Quacks are people without any level of- any form of pharmaceutical training or any formal training. Maybe they were um, employed in a pharmacy . . .and then they've seen 'oh, this is a good business, it's about just giving drugs when people come!' and they set up their own pharmacy outlets. They may employ [pay] a registered professional who lends them their license. But the quack runs the show."–Pharmaceutical Society of Kenya board member

> "Especially at the Coast, it's very porous. They [drugs] come from Tanzania through unofficial routes–in buses, in suitcases. It's only the authorized medicines that should be in the market. . . Especially with the family planning commodities–we've had *problems*. Either they are counterfeit or falsified products, especially Postinor-2 [an emergency contraceptive pill]—we've received so many complaints."–Pharmacy and Poisons Board regulator

Regulators indicated that there were existing mechanisms for increasing accountability, but that these often worked better in theory than practice and did not necessarily deter illegal activity. For example, the Pharmacy and Poisons Board had implemented an SMS-based pharmacy registration system with the goal of providing community members with a tool to verify the registration status, licensed owner, and location of the pharmacy they visited. However, when asked how the system functioned in practice, one respondent noted:

> "People are using it but it's not up to what we were expecting initially, despite the fact that it is free of charge. . .It's only for the licensed [pharmacies]. Because, for the unlicensed, number one, they should not be there *(laughs)*, then secondly, uh, you don't have much control."–Pharmacy and Poisons Board regulator

## Suggestions to improve pharmacy service quality and regulation compliance

Study participants had suggestions to improve service quality and regulation compliance in private pharmacies. One was educating consumers as to what they should look for when seeking quality pharmacy services and empower them to demand it.

> "We advise them [customers] to use drugs that are registered because there are some drugs that are from black market, so we advise them to [visit] a registered shop, ask for drugs that you can confirm if they are original drugs."–Trained pharmacist or pharmaceutical technologist (not specified)

> "I would like the chemist to keep papers to show that this is a doctor or a chemist attendant. So that we can avoid those who act to be doctors and they are not."–Young female contraception purchaser (injection)

Participants also indicated a desire to see regulatory mechanisms strengthened. In particular, county and national-level stakeholders and policies routinely referenced need for an overhaul of the country's Pharmacy and Poisons Act, expressing frustration that Act remained the source legislation for present-day pharmacy practice in Kenya, despite its not having kept pace with decades worth of changes in evidence and pharmacy practice. Respondents also speculated that the Act's punitive measures on unregistered pharmacies would hardly be considered a deterrent in modern-day Kenya.

Additionally, participants identified a need improve opportunities for professionals working in retail pharmacies to participate in CPD and other training opportunities. They noted that trainings usually only actively targeted private pharmacy personnel when driven by a non-governmental organization, external donor initiative, or pharmaceutical company.

"There is this organization called [international NGO] that has been empowering us, they call us to seminars and empower us on that, so then we are able to help the community with that information."–Pharmacist

Participants encouraged more of these strategic collaborations to reach private sector pharmacy personnel, in order to both reinforce existing skills but also expand their scope of authorized activities, including contraception provision activities.

"For example, in the past we were unable to test for malaria in the chemist, but we were taken for training and nowadays we test. If donors would come and say we also want pharmacists to help [with family planning] injections, they help us. So that if someone comes and wants to be injected we can help such a client."–Trained pharmacist or pharmaceutical technologist (not specified)

Finally, a board member of the Pharmaceutical Society of Kenya suggested 'branding' pharmacies, that is, developing a visible and recognized mark of quality which pharmacies could join through demonstrated adherence to a set of standards. Such an exercise would require collaboration across governmental, non-governmental, and possibly private stakeholders. Incentives to join would include opportunities for training and the ability to provide an expanded set of services (as suggested above).

"Let's say I own a pharmacy, my own personal pharmacy–I would want to train my staff on certain things: customer service, even pharmaceutical knowledge just to refresh their knowledge. We have partners who are ready to do that, so when we offer those kinds of [trainings], we believe a lot of pharmacists will want to join the chaining and branding initiative. And then of course because of the standards we'll have for joining, if you're a quack you won't join. You'll have to be a registered and licensed professional to join. And in this proposal we're proposing expanded services at the pharmacies. So we need approval from–we need buy-in from the Ministry of Health."–Pharmaceutical Society of Kenya board member

## Discussion

This study described a private retail pharmacy sector dispensing contraception with its own momentum, its regulators a few steps behind. Specifically for injectable contraception, in Kwale County under the current regulatory policy, some pharmacies had become 'one-stop-shops' when it came to injectable contraception. Clients wanting a contraceptive injection could often purchase the contraceptive and be injected on site, either through strategic collaborations with other health system personnel, or through pharmacy staff expanding their scope of services on their own (without authorization). These models are common around the world. Pharmacies are an established source for injectable contraception across Africa and Asia [3, 7] and nurses, doctors, and other clinical officers have also been observed working as drug-dispensing (and administering) staff elsewhere in the region[23], though there is a lack of

evidence from LMICs evaluating these multidisciplinary collaborations between pharmacy personnel and other medical personnel[8]. Similar to reports from this study of pharmacy personnel injecting clients, pharmacy personnel in Uganda have also reported providing injections themselves, without official authorization[24].

Regulators and young community members using pharmacies described the sector as being difficult to monitor, support, empower and track. This had a deleterious effect on the profession's reputation, including among the young people who relied on pharmacies as a source for contraception services. Our young community participants—even those who regularly purchased contraception from pharmacies—were all aware that among the private pharmacies in Kwale County, there were unregistered establishments with unqualified personnel dispensing poor-quality commodities. Young people frequently cited uneducated pharmacy personnel as a drawback to accessing care at pharmacies and did not feel able to identify unqualified personnel themselves. Their concerns mirror findings elsewhere in the literature. One systematic review on the quality of private pharmacy services highlighted similar shortcomings across 30 studies in 15 countries relating to dispensing of appropriate medicines in correct dosages, a lack of qualified pharmacy personnel doing the dispensing, and varying quality in advice given to clients, if it was given at all.[9] Another systematic review looking at practices of 'specialized drug shops' in sub-Saharan Africa found 62 articles describing similar variance in pharmacy personnel's training, their knowledge, and dispensing practices.[25]

Pharmacy services in Kenya have received increasing attention over the last few years. There have been recent, controversial efforts to revise the Pharmacy and Poisons Act (CAP 244) [26], and proposals to create a new regulatory agency, the Kenya Food and Drug Authority [27]. Additionally, in 2017 Kenya's president named universal health coverage as one of the country's four key development priorities. In this context, there is renewed interest to link with private sector healthcare providers in order to improve the reach of health services beyond what the public sector can offer[28]. Now is the time to propose measures to improve the quality of pharmacy services and personnel. Participants in this study had several ideas in this regard.

Participants suggested measures to increase clients' awareness of and demand for regulated, quality pharmacy services. There is existing client-side awareness of the need for quality services: the concerns expressed by this study's participants are corroborated by recurring news coverage of illegal pharmacies, unqualified pharmacists, and counterfeit medicines[21]. The existing SMS based system for clients to check the registration and ownership status for their neighborhood pharmacy can help to improve client-side accountability. A similar proposed system to improve the quality, safety, and effective use of medicines available in Kenya would also apply unique codes to any imported and locally-manufactured medicines, so that clients could obtain SMS-based information on whether the drug was real and registered, as well as its side effects[29]. However, Kenya's Pharmacy and Poisons Board will need adequate resources to ensure that pharmacy clients around the country are sensitized to these services and empowered to use them.

Participants also emphasized a need for improved government regulations, as well as improved access to CPD and other in-service training for retail pharmacy personnel. For contraceptive injections in particular, respondents recognized the opportunity to officially expand access of injectable provision by training pharmacy personnel to both dispense and inject. While this proposal is out of line with the with Kenya's current family planning guidance, a 2017 family planning task sharing brief from the World Health Organization indicated that pharmacists, and even pharmacy workers in certain circumstances, could administer injectable contraceptive services, provided they received appropriate training[30]. Additionally, the very recent advent of self-administrable injectable contraception (DMPA-SC) and growing

evidence that self-administration can match or improve rates of injection continuation as compared with provider administration [31], presents another possible role for pharmacy personnel: that of counsellor and dispenser, with pharmacies offering women a clean, discreet place to self-inject and safely dispose of their injectable.

Finally, the suggestion of a branding exercise to create a network of 'quality' pharmacies, brings together all of the above and describes a multi-faceted approach to improving pharmacies as contraception providers: client- and provider-focused interventions as well as improved regulatory capacity. The need to *holistically* change pharmacy practice has been recognized elsewhere in the literature [23, 32, 33], notably in neighboring Tanzania. For its lower-tier drug shops, Tanzania's accredited drug dispensing outlet (ADDO) program implemented a series of interventions to improve public-sector regulatory capacity; educate the public on pharmacy service quality and treatment compliance; train and supervise staff on dispensing skills; train drug shop owners on business skills; and provide business incentives for achieving ADDO accreditation (for example, access to microfinancing, or the ability to sell an expanded set of essential medicine)[34]. Similar accreditation and social franchising programs have been launched in Liberia and Uganda [34, 35], Thailand and Vietnam[32], and are being explored in Bangladesh [3, 36]. Accreditation programs combine business incentives with public sector standards-setting and may provide a rich potential ground for collaboration between Kenyan public sector regulatory agencies; educational institutions; professional associations; drug distributors; and individual retail pharmacies.

This study had some limitations. Our mystery shopper could only inquire as to whether she could receive a contraceptive injection on site, so in those cases where she was told on-site injection was possible, she could not confirm *who* would have administered it and had to rely on what she was told by the pharmacy personnel. As our mystery shopper was meant to reflect likely actions of a nervous young client, she was also not able to ask for the training or position of the person who assisted her. Our youth participants in focus group discussions may have felt uncomfortable discussing contraception in a group environment; we did our best to mitigate any discomfort by using a series of vignettes for participants to react to and having our data collectors keep any discussion from lingering on personal experiences. Additionally, during interviews with pharmacy personnel, the participant's background and training often came up in discussion; however, in an oversight, we did not systematically confirm training in all interviews, so there were some participants for whom this information was missing. Social desirability bias may have also caused certain interviewed pharmacy personnel to not report their dispensing practices accurately, especially if they knew they were out of line with regulations. However, a major strength of this study was the use of multiple methods and inclusion of both young clients and pharmacy personnel to triangulate actual pharmaceutical practices from the various reported practices.

## Conclusion

Pharmacy personnel (trained and untrained) around the world serve as clients' primary point of contact with the health system. As observed in this study focusing on injectable contraception provision to young people, retail pharmacies have expanded to contraception service provision, and pharmacy personnel's roles have transcended formal or informal training previously received. However, Kenya like many other countries, faces challenges in monitoring the types of services available in pharmacies, the quality with which they are provided, and who does the providing. Public health initiatives, including those related to contraception, often struggle to engage with private providers who fall out of the public healthcare system. That said, investing in multi-pronged initiatives with both public and private partners to

improve overall pharmacy practice, update and enforce regulations, and educate the public can strengthen services at young people's preferred point of care.

## Supporting information

**S1 File. Focus Group Discussion guide.**
(DOCX)

**S2 File. In-Depth Interview guide for young contraception purchasers.**
(DOCX)

**S3 File. Key Informant Interview guide (for a person working in a pharmacy).**
(DOCX)

**S4 File. Mystery injection shopper checklist.**
(PDF)

## Acknowledgments

We appreciate the support of Jefferson Mwaisaka and Winnie Wangari. This work was partially supported by the UNDP/UNFPA/UNICEF/WHO/World Bank Special Programme of Research, Development and Research Training in Human Reproduction (HRP). The manuscript represents the view of the named authors only.

## Author Contributions

**Conceptualization:** Lianne Gonsalves, Adriane Martin Hilber.

**Formal analysis:** Lianne Gonsalves.

**Funding acquisition:** Lale Say.

**Investigation:** Lianne Gonsalves, Peter Gichangi.

**Methodology:** Lianne Gonsalves, Adriane Martin Hilber.

**Project administration:** Peter Gichangi.

**Supervision:** Kaspar Wyss, Peter Gichangi, Adriane Martin Hilber.

**Writing – original draft:** Lianne Gonsalves.

**Writing – review & editing:** Kaspar Wyss, Peter Gichangi, Lale Say, Adriane Martin Hilber.

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
