## [Decision Letter · Decision Letter 0]

21 Nov 2019

Regulating pharmacists as contraception providers: A case study from Coastal Kenya on injectable contraception provision to youth

PONE-D-19-21933

Dear Dr. Gonsalves,

We are pleased to inform you that your manuscript has been judged scientifically suitable for publication and will be formally accepted for publication once it complies with all outstanding technical requirements.

With kind regards,

Mellissa H Withers, PhD, MHS

Academic Editor

PLOS ONE

Journal Requirements:

1. Please include a copy of the interview guide used in the study, in both the original language and English, as Supporting Information, or include a citation if it has been published previously.

2. We would suggest a slight alteration to the title to remove 'case study' and include 'qualitative study' as follows: "Regulating pharmacists as contraception providers: A qualitative study from Coastal Kenya on injectable contraception provision to youth"

If this is OK, please incorporate this change in the final manuscript files and we will update the online submission system.

Reviewers' comments:

Reviewer's Responses to Questions

**Comments to the Author**

1. Is the manuscript technically sound, and do the data support the conclusions?

Reviewer #1: Yes

2. Has the statistical analysis been performed appropriately and rigorously? 

Reviewer #1: Yes

3. Have the authors made all data underlying the findings in their manuscript fully available?

Reviewer #1: Yes

4. Is the manuscript presented in an intelligible fashion and written in standard English?

Reviewer #1: Yes

5. Review Comments to the Author

Reviewer #1: In this qualitative study, the authors evaluate the availability and quality of services for injectable contraception for young women in Kenya. Using focus groups, in depth interviews, key informant interviews, and a mystery shopper, the authors were able to look at the question from many angles and utilized the study participants to imagine improvement strategies. While this study is specific to the Kenyan context, the authors put the information in the context of research in this field globally. They also did an excellent job of discussing best practices and WHO guidelines and recommendation as they discussed improvement opportunities.

The writing was overall excellent.

In the abstract (Page 2, lines 41-42), the separate thoughts should be separated by commas rather than semicolons. With these exceptions, I did not see any grammatical errors.

This qualitative study is a valuable contribution to the literature and I commend the authors on a job well done.

6. PLOS authors have the option to publish the peer review history of their article (what does this mean?). If published, this will include your full peer review and any attached files.

Reviewer #1: No

---

## [Editor Report · Acceptance letter]

10 Dec 2019

PONE-D-19-21933 

Regulating pharmacists as contraception providers: A qualitative study from Coastal Kenya on injectable contraception provision to youth

Dear Dr. Gonsalves:

I am pleased to inform you that your manuscript has been deemed suitable for publication in PLOS ONE. Congratulations! Your manuscript is now with our production department. 

With kind regards,

on behalf of

Dr. Mellissa H Withers 

Academic Editor

PLOS ONE